# Chemical Diversity between Three Graminoid Plants Found in Western Kenya Analyzed by Headspace Solid-Phase Microextraction Gas Chromatography–Mass Spectrometry (HS-SPME-GC-MS)

**DOI:** 10.3390/plants10112423

**Published:** 2021-11-10

**Authors:** Linus Svenberg, Åsa Emmer

**Affiliations:** Analytical Chemistry, Division of Applied Physical Chemistry, Department of Chemistry, School of Engineering Sciences in Chemistry, Biotechnology and Health, KTH Royal Institute of Technology, Teknikringen 36, SE-100 44 Stockholm, Sweden; aae@kth.se

**Keywords:** *Cynodon dactylon*, *Cyperus exaltatus*, *Panicum repens*, HS-SPME-GC-MS, volatile profile

## Abstract

In recent work, it was shown that the graminoid plants *Cynodon dactylon* (*Poaceae*), *Cyperus exaltatus* (*Cyperaceae*), and *Panicum repens* (*Poaceae*) have an ovipositional effect on the malaria vector *Anopheles gambiae* in olfactometric bioassays. In order to get a view of the diversity of semiochemicals present in the environment of the vector during olfactometric trials, in the present work, the volatile profiles of these graminoid plants were analyzed using headspace solid-phase microextraction (HS-SPME) together with gas chromatography–mass spectrometry (GC-MS). In addition, one-way ANOVA comparison of compounds detected in two or more headspace samples are presented in order to provide a basis for comparison of compounds that could constitute a starting point for novel blends of volatile organic compounds to be tested as oviposition attractants.

## 1. Introduction

The study of the influence of volatile compounds and their behavioral effects on insects is an important aspect of chemical ecology. The development of control strategies against disease vectors, based on the semiochemicals available from plants, has, for example, led to advances in new pest control concepts. One goal is to decrease the spread of diseases associated with certain species of insects. An example of such a severe problem is the spread of malaria in sub-Saharan Africa, which has declined [1] when compared to the goals set by the World Health Organization (WHO) in their Global Technical Strategy for Malaria (GTS) report from 2016 [2]. As of now, long-lasting insecticide nets (LLINs) and indoor residual spray (IRS) are the most commonly employed techniques to minimize the infection rate of malaria. However, these methods are limited to indoor use, and it has been shown that the effectiveness has subsided due to a decrease in usage [1]. Increased resistance in certain *Anopheles* mosquitoes towards the utilized insecticides has also been reported [3]. One of the main malaria vectors is the *Anopheles gambiae* mosquito, whose host-seeking [4,5] and oviposition behaviors [4,5,6] are impacted by the semiochemicals in its environment. In order to obtain new methods of vector control, the regulation of oviposition using semiochemicals has thus been suggested, focusing on outdoor settings, such as “lure-and-kill”, that can provide an alternative to the indoor methods [7,8]. The challenge in the development of these novel outdoor-focused control techniques is the identification of volatile organic compounds (VOCs) that could be used to influence the malaria vectors’ choice of egg-laying site.

To be able to determine the volatile compounds that elicit this behavior from the target disease vectors, the presence of the VOCs must be established. With the knowledge of what VOCs are present in the chemical environment of the vector as oviposition occurs, one can aim to relate the VOCs to the behavior of the vector. In addition to establishing VOCs in the chemical environment, the sources of these VOCs should be determined. In a study by Bokore et al., a correlation between graminoid plants and the occurrence of *A. gambiae* instars was shown. It was determined that the choice of egg laying sites is influenced by the presence of water and certain graminoid plants. From this study, it could also be seen that the plants associated with the occurrence of the instars of *A. gambiae* are *Cyperus rotundus* (Nut grass) and *Cyperus exaltatus* (Exaltatus Grass) from the Cyperaceae family, as well as *Cynodon dactylon* (Bermuda Grass) and *Panicum repens* (Torpedo Grass) from the Poaceae family [9]. The essential oil composition [10,11,12] of *Cyperus rotundus* has previously been studied, as well as the volatile headspace of its macerated rhizomes using headspace solid-phase microextraction (HS-SPME) [13]. As the volatile constituents of Nut grass has been mapped previously [14], and the fact that this grass has been studied for its ovipositional impact on the malaria vector *A. gambiae* [15], it was omitted in the present study.

Analysis of some graminoid plants has been performed previously, but this has been done on dried hay infusions of the Bermuda grass [16]. The geographical origin of a plant has been suggested to be an influencing factor in the composition of the volatiles and essential oil [17,18]. Furthermore, this was demonstrated in a comparison of the essential oil of rhizomes of Nut grass from two different locations in South Africa [19], adding to the incentive of determining the volatile profile of the graminoid plants from the specific location in Western Kenya suggested by Bokore et al. [9].

In addition, the olfactometric work presented by Bokore et al. shows that even uprooted plants still elicit some ovipositional effects on gravid *A. gambiae* [20]. In the study by Bokore et al., dynamic headspace analysis was reported for the three graminoid plants used in the present work. The composition of volatiles released from the uprooted grass shoots under the two-port olfactometric bioassay circumstances was studied. However, headspace analysis of the shoots and root parts of these plants separately has not been studied. Thus, there is a lack of insight into the auxiliary compounds that are present, and that potentially contribute to the ovipositional effect on the malaria vector in the conditions used in the previous study [20].

In the present study, headspace (HS) sampling was used in combination with solid-phase microextraction (SPME), as this offers a rapid, solvent-free, and qualitative analysis. The application of SPME in the study of gas phase composition is common practice today and is applied in the investigation of flora samples for the determination of chemical composition [21,22,23].

Here, HS-SPME GC-MS analysis was performed on shoots and root parts from the three graminoid plants: *Cynodon dactylon*, *Cyperus exaltatus*, and *Panicum repens*, to determine the composition of volatile compounds acquired from each plant in similar conditions to that of the two-port olfactometric setup [20]. In addition to this, the data collected were interpreted with one-way ANOVA and Tukey post-hoc test to determine if there was a significant difference in the relative composition between the three graminoid plants, and between the different parts of the plants under conditions emulating certain aspect of those described by Bokore et al. [20].

## 2. Results and Discussion

### 2.1. Chemical Composition of Headspace Samples from C. dactylon, C. exaltatus, and P. repens

In Table 1, the 46 compounds that were identified in the roots and/or the shoots of the three different graminoid plants are listed. The compounds were detected and tentatively identified using MS as well as experimental retention index (RI). External analytical standards were used to confirm the identity of the compounds when possible. The compounds found were categorized into one of the following classes: terpenoids (TR), aliphatics (AL), benzenoids and phenylpropanoids (BP), C5-branched compounds (C5), nitrogen- or sulfur-containing compounds (NS), and cyclic miscellaneous compounds (Cyc) [24]. These chemical classes represent the continuation ofthe initial work by Knudsen et al. [25]. The categorization of compounds was based on the results from HS sampling of plants and covers over 1700 compounds [24]. Thus, it is appropriate to use these classes as many compounds have already been categorized accordingly. It should be noted that in the work by Knudsen et al. [24], 7 classes were used to classify the compounds. In this work, nitrogen- and sulfur-containing compounds were grouped under the same term, giving a total of 6 classes. In addition, a second classification method was used to classify the compounds based on their functional group, which can be seen in Table 1.

In order to compare the results of the analysis of the headspace of the graminoid plants, the first task was to identify the source tissue of the different compounds. In Figure 1A, the number of compounds detected in the root samples, shoot samples as well as in both types of samples are shown. Here, the largest number of compounds detected in both the root and shoot samples were obtained for *C. dactylon*. On the other hand, *C exaltatus* showed the largest total number of unique compounds detected in the root and shoot samples, respectively. Regarding *P. repens*, all the compounds found in the root samples were also found in the samples of the shoots. Finally, even though VOCs can be emitted by almost any plant tissue [26], it can be noted that the largest number of volatiles were identified in the samples from the shoots in all three species. After grouping the compounds under one of the 6 classes mentioned previously, it could be established that out of the 46 compounds found, 31 were terpenoids, 8 were aliphatic, 4 were cyclic compounds, 2 were a benzenoid/phenylpropanoid, and 1 was a nitrogen- or sulfur-containing compound. No C5-branched compounds were detected in the headspace samples. The distribution of the number of compounds in each of the classes, derived from the analysis of the three graminoid plants, follows the expected trends and is shown in Figure 1B. It is expected that the number of terpenoids, i.e., mono-and sesquiterpenes, should constitute the largest percentile of the identified compounds [27]. It is also more common that the number of fatty acid derivatives, such as the larger aliphatic compounds, appear in higher numbers than compounds containing a benzene ring [27]. Figure 1C shows how many of the compounds were detected for each of the graminoid plants as well as the number of compounds overlapping in two or more plant samples.

In Figure 2, the distribution of grouped compounds found in each of the plant species is shown separately. In Figure 2A, it can be seen that the highest number of the dominating group of terpenoids could be found in the root samples of *C. exaltatus*, when compared to the root samples of the other two plant species. The lowest total numbers of detected peaks were obtained from the root samples from *P. repens*. Nonetheless, in Table 1, it can be seen that the largest relative peak area was detected in *P. repens* across all samples. It appears that fewer VOCs originated from the torpedo grass roots compared to the other plant samples but that the roots emitted an amount represented by a higher relative area percentage of the said compounds. In Figure 2B, the distribution between substance classes detected in the shoot samples is shown. Here, it is visible that *C. exaltatus* shoot samples contained the largest number of terpenoids of all the samples analyzed, contributing to the trend of the terpenoid class constituting the largest percentile in the overall class distribution. In Figure 1C, the Venn diagram shows the overlap of the detected compounds in the different plants. From this figure, it can be observed that the number of common compounds found between the three plants are quite similar. Six compounds were found in all three plants, while one to three compounds were shared between pairs of plants. What is apparent in Figure 1A,C is the large number of compounds detected only in the headspace of the *C. exaltatus* samples. The larger number of compounds detected for this plant could be the basis for distinguishing the plants by family, as the more fragrant *C. exaltatus* belongs to the *Cyperaceae* family. Both *C. dactylon* and *P. repens* belong to the Poaceae family of plants, and when compared to that of *C. exaltatus*, they have a similarly low number of uniquely detected compounds in their headspace. Although grouping of plant families based on VOCs emission has been performed [28], it has not been done for the specific plants analyzed in this work. Therefore, differences and similarities based on belonging to a family can be suggested but not determined.

In total, 26 compounds were detected for *C. exaltatus* alone, where β-elemene and cyperene showed the highest relative area. In the headspace of *C. dactylon*, six compounds, the highest relative areas are reported for germacrene D and mesitylene, were uniquely detected. Regarding *P. repens*, only two compounds, myrtanylamine and phenylethyl alcohol, were uniquely detected. In the case of myrtanylamine, the sparse information available in the literature regarding its presence in the headspace of plants should be noted. This shows an excerpt of VOCs that separates the chemical profile of the plants from each other, and that could be a factor in why these plant species show different ovipositional strength [20]. Understanding the diversity of the compounds originating from all the graminoid plants is important for the selection of compounds to be tested as oviposition attractants. However, it is also important to investigate compounds that are shared between the headspace samples of multiple plants, and to evaluate the differences in the abundance between different plants. Compounds, such as α-pinene, β-pinene, and limonene, have been shown to elicit antennal responses from *A. arabiensis* [29]. Moreover, it has been shown that a synthetic blend of compounds, with a naturally observed ratio, could trigger a short-range response from *A. arabiensis* at a certain dosage [29]. Nonetheless, the same synthetic blend showed no response at higher dosages, or even provoked an avoiding response from the mosquitoes [29]. It has been suggested that compound blends are significant in the development of ovipositional attractive odors, and that not only must the individual compounds be determined, but also the ratio at which they occur in relation to each other [30,31]. This would suggest that a different response could be elicited for a blend of the same compounds with different ratios. Thus, compounds that appear in several plant samples could induce a positive ovipositional response, if the ratio of the compounds is suitable for the specific target. On the other hand, the response could vary between plant species due to the different abundances of the same compound. Therefore, it is valuable to determine if there is a significant difference in the relative peak area of the detected compounds between the different plants.

From the Venn diagram in Figure 1C, it can be seen that there are six compounds common to the headspace samples of all three species when the results from the roots and shoots are combined. These compounds are α-pinene, β-pinene, 3-octanone, β-myrcene, limonene, and citronellol. Three compounds were shared between *C. dactylon* and *C. exaltatus*: 2-pentyl-furan, nonanal, and 2,2,6-trimethyl-cyclohexanone, while *C. dactylon* and *P. repens* only shared two compounds, namely decane and β-ionone. The only compound shared by *C. exaltatus* and *P. repens* was 3-carene. Since the results are also split into roots and shoot samples, a similar comparison can be made for compounds common in both parts of the plants, to investigate if there are significant differences in the abundance of VOCs in the different plant parts. It has been suggested that compounds emitted into the rhizosphere play a role in plant–plant interaction [32], as well as plant–herbivore interactions [33]. However, the interaction of rhizosphere compounds and the malaria vector has not been fully explored, with regards to whether they contribute to long-range cues or short-range cues. Knowledge of abundance variation could offer additional insight into which compounds are important for oviposition. For *C. dactylon*, the compounds α-pinene, β-pinene, 3-octanone, decane, and limonene were present in both root and shoot samples. In the same sense, the compounds β-pinene, cyperene, and β-selinene were present in the samples of *C. exaltatus*. For *P. repens*, the compounds β-pinene and 3-octanone were present.

### 2.2. Statistical Evaluation of Results

The present study does not provide a concentration-based chemical profile, but rather, the relative peak areas were utilized. Nonetheless, determination of the significant differences in the detected VOCs is important to obtain an understanding of the sources that influence the chemical environment of the vector, which could affect the behavioral choices, as mentioned previously. Thus, one-way ANOVA was applied to identify differences in the composition and their significance. When applying one-way ANOVA, homoscedasticity, which is equal variance, and normality in the dataset are assumed for the robustness of the analysis. In order to obtain this, there has to be a repetitive detection of the compounds that can be analyzed with ANOVA.

Evaluation of the differences in the abundance of the detected compounds, and the significance of these differences, between plants is important to understand the role of a VOC as a potential attractant. Out of the compounds mentioned previously, the analysis could not be applied to the compounds α-pinene, limonene, decane, β-Ionone, 2-pentyl-furan, nonanal, and 2,2,6-trimethyl-cyclohexanone as these compounds were not detected in a repetitive fashion, to assume homoscedasticity and normality of the data. Therefore, one-way ANOVA was applied to all three plant species for the compounds β-pinene and 3-octanone, for α-pinene and citronellol when comparing *C. dactylon* with *P. repens*, and β-myrcene when comparing *C. exaltatus* with *P. repens*. After the one-way ANOVA analysis was performed, the post-hoc Tukey test was also performed in order to identify if any significant difference in the mean relative peak area could be found between any of the plants (Table 2).

Similarly, the evaluation of differences was done for the previously mentioned compounds comparing the roots and shoots of the plants (Table 3).

Here, it can be seen that there were no significant differences in the detected relative peak areas except for the peak areas of β-pinene in the *P. repens* sample when compared to the other two plants. Furthermore, in Table 1, it can be seen that the relative peak area for β-pinene is very large when compared to the other compounds detected in the headspace of the torpedo grass. A tendency towards this could also be discerned when comparing the percentages in the other plants. The results shown in this study suggest that there are very small differences in the normalized amount detected in the analysis of the overlapping compounds, which were compared based on the criteria described earlier. With the exception of β-pinene, the relative peak areas were not significantly different in the 95% confidence interval for any of the compounds. Furthermore, it can be seen in Table 3 that when comparing the compounds found in both the shoots and the roots of the different plants, no significant difference could be seen for the normalized peak area for any of the compounds listed for *C. dactylon.* Similarly, no significant difference could be shown for the normalized peak area of β-selinene in *C. exaltatus* or for 3-octanone in *P. repens*. A significant difference could be observed for β-pinene in both *C. exaltatus* and *P. repens*, where the normalized peak area of β-pinene was significantly larger in the root tissue samples than in the shoot samples. The same result was obtained for cyperene in the comparison of the roots and shoots for *C. exaltatus*.

### 2.3. Relating Findings to the Olfactometric Results

Since β-pinene stands out in the obtained results, it is of interest to discuss this component’s possible response effects. It was reported earlier [29] that β-pinene elicits an antennal response from *A. arabiensis*, which would suggest that a similar response from *A. gambiae* is plausible as these mosquitos belong to the same family. From the work presented by Bokore et al. [20], it was suggested that in the short-range olfactometry tests, all three graminoid plants generated a similar preference from the malaria vector when compared to lake water. However, during long-range attraction trials, *P. repens* showed the weakest attraction when compared to lake water. Relating these olfactory measurements to the results reported here, regarding the occurrence of β-pinene is not straight-forward. It could be argued that the reduced attraction of *P. repens* at long range could be a result of β-pinene showing a significantly higher abundance in the roots, which were more exposed in short-range experiments, than in the shoots of the plant. This could suggest the role of β-pinene as a short-range cue, while not acting as a long-range cue. *C. exaltatus* showed similar attraction effects as *C. dactylon*, despite the fact that a significantly larger normalized peak was detected for β-pinene in the root sample of *C. exaltatus*. A “blend” effect from the combined compounds present in the headspace of the plants is possible as discussed in Section 2.1. In order to move forward with identifying new attractants for the mosquitoes, using the blends of compounds reported for *C. dactylon* and/or *C. exaltatus* in Table 3 could be a starting point. This choice is motivated by the fact that it was reported that *P. repens* induced a lower attraction at long range, while the other two graminoid plants were reported to show both short- and long-range attraction. Still, no significant difference in the abundance of the compounds common to these two plants could be elucidated. This suggests that investigating different ratios of the compounds that are common for the roots and shoots of the plants should be considered for the testing of new blends. These compounds could be involved in both the short- and long-range attraction of the vector, suggesting their potential as “blend-effect” chemicals. After these blends are evaluated, the addition of individual compounds to the blend could be investigated for potential increased effect on the malaria vector. Examples of such compounds could be β-elemene, which was detected in *C. exaltatus* shoot samples, or germacrene D, which was only detected in the root samples of *C. dactylon*. However, the addition of single compounds will be more of a “hit-and-miss” approach, although the tentative selection of compounds has been narrowed down with the results reported in this study.

## 3. Materials and Methods

Grass samples were collected in the summer of 2020 and shipped to KTH Royal Institute of Technology, Stockholm, Sweden. *C. exaltatus* was collected from Rusinga Island (0°23′47.3″ S 34°12′13.1″ E) while *C. dactylon* and *P. repens* were collected in Mbita (0°26′06.19″ S 34°12′53.13″ E). The plants were divided into roots and shoots prior to import due to permits from the Swedish Board of Agriculture. Roots include any plant tissue that is present below the ground, while shoots refer to any part of the plant tissue that is above the ground. Botanical identification was performed at the International Centre of Insect Physiology and Ecology Thomas Odhiambo Mbita Campus (ICIPE) in Kenya. Upon arrival, both parts (shoots and roots) of the grass were washed in order to remove any residual soil left on the plant material. The plant material was washed using doubly distilled water (Milli-Q) from a Synergy 185 water purification system (Merck, Kenilworth, NJ, USA) with a resistivity of 18.2 MΩ*cm at 25 °C. Excess water from the washing of the plant material was then allowed to evaporate at room temperature from the samples. The samples were then stored in a −80 °C freezer until further sample preparation and analysis.

### 3.1. Solid-Phase Micro Extraction

In total, 10 g of sample were placed in a 250 mL round-bottom flask, covered with 100 mL of Milli-Q water, and sealed. The water-grass mixture was allowed to sit at room temperature for 30 min before the SPME fiber was introduced to the round-bottom flask neck. A picture showing a sample collection setup is provided in the Appendix A. The fiber was exposed to the headspace for 4 h. The SPME fibers used were 1 cm Polydimethylsiloxane/Divinylbenzene (PDMS/DVB) fibers with a 24 gauge needle (57310-U, Supelco, Bellefonte, PA, USA). Each shoot and root headspace sample collection was performed with two different samples on two different days (day 1 and 2), due to the long extraction time. Two different fibers were used for sample extraction on day 1 and 2, so that the total number of analyses for each root and shoot sample was 4. The fibers were conditioned prior to sample and blank analyses by being placed in the injector port of the GC instrument for 3 min, while the temperature in the injector was 40 °C for 6 s, then ramped to 260 °C at a rate of 12 °C/s, and then held for 3 min. The resulting chromatograms were studied for any contaminations and undesired compounds to determine if the fiber was ready to be utilized for sample extraction. After extraction, the fiber was withdrawn, injected into the GC injector port, and the GC method was started. A desorption time of 30 s in the port was allowed before removing the fiber. The sample collection method was based on the work by Svenberg et al. [34].

### 3.2. Gas Chromatography–Mass Spectrometry Parameters

Analysis was performed using an Agilent 7890A GC (Agilent Technologies, Santa Clara, CA, USA) coupled to a 5975C Triple axis MS (Agilent Technologies). The temperature program started at 40 °C and was held for 1 min, after which the temperature was ramped to 260 °C at 10 °C/min. After reaching 260 °C, the temperature was held for 5 min. A 30 m × 0.25 mm × 0.25 µm DB-5 column (Agilent Technologies) was used for all analyses. The temperature program for the inlet was the same as the program described in the previous section. The carrier gas used was 6.0 LAB LINE helium (Strandmöllen AB, Ljungby, Sweden) and the mass range of the 5975C MS was set to 35–400 *m/z*. All data handling was performed and exported using the *Data analysis* software in *Chemstation* (Agilent Technologies), and identification with mass spectrometry was done with the National Institute of Standards and Technology (NIST) MS Search 2.0 program for the NIST/Environmental Protection Agency (EPA)/National Institute of Health (NIH) Mass Spectral Library version 2.0 g, build 2009.

Peak detection was performed by using the integration function in *Chemstation* data analysis software. In the method for peak detection, the parameters were set to initial area reject of 1,500,000, initial peak width of 0.02, shoulder detection off, and initial threshold of 16.0. For determination of the total area in the chromatogram, the parameters of the integration method were set to initial area reject of 1, initial peak width of 0.02, shoulder detection off, and initial threshold of 15.0. Retention indices were calculated by comparison of the retention times to that of the 49452-U C_7_-C_40_ alkane standard (Supelco, Bellefonte, PA, USA) (10 µg of each in hexane), and external standard identification was performed by comparing the sample analyte peak retention times to that of the retention times of compounds present in the CRM40755 Cannabis Terpene Mix A (Sigma Aldrich) (10 µg of each in hexane). The mix contained the following 20 terpenes: α-pinene, β-pinene, camphene, 3-carene, α-terpenine, R-(+)-limonene, γ-terpinene, L-(−)-fenchone, fenchol, (1R)-(+)-camphor, isoborneol, menthol, citronellol, (+)-pulegone, geranyl acetate, α-cedrene, α-humulene, nerolidol, (+)-cedrol, and (−)-α-bisabolol. The standards of β-caryophyllene standard 22075 (Sigma-Aldrich) and the (−)-caryophyllene oxide 91034 (Sigma-Aldrich) were also used to confirm the identity of the detected compounds. Analysis parameters, peak detection parameters and peak selection were based on the work by Svenberg et al. [34].

All statistical analysis and graphical work were performed in the Rstudios R software. Source code, libraries required, and data tables are provided in the Appendix A.

## 4. Conclusions

In the present work, 46 volatiles were detected and identified in the root and/or shoot parts of the three graminoid plant species: *C. dactylon*, *C. exaltatus*, and *P. repens*. It was shown that out of the 46 detected compounds, 26 were unique to *C. exaltatus*, 6 were unique to *C. dactylon*, and 2 were unique to *P. repens*, with 12 compounds overlapping in combinations between the studied plants. Furthermore, it was shown, with one-way ANOVA analysis and Tukey post-hoc test, that there was no significant difference in the total relative peak areas detected in the shoot and root samples for any of the overlapping compounds except for β-pinene. The relative peak area for this compound was significantly higher in *P. repens* compared to the other two plants, while no significant difference was shown for this substance between *C. dactylon* and *C. exaltatus*. The comparison of *C. dactylon* and *C. exaltatus* showed that for the overlapping compounds, there is no significant difference in the relative area of the compounds detected in this study. In addition to this, it was shown that when comparing the relative area of compounds detected in both the root and shoots of the different plants, there are significant difference in the areas detected in this work. The knowledge about these compounds appearing both in the roots and shoots, in combination with the previously shown attraction strength of the different grasses, offers new starting points for blends of compounds. Blends of compounds including, e.g., β-pinene, cyperene, and β-selinene could be possible starting points, based on data reported for *C. exaltatus* in this work. These blends have to be investigated further to determine the ratio that should be used to elicit a response from the vector. Once this has been determined, the individual unique compounds found in the present study, such as β-elemene, can be added to the proposed plant-specific blends.

## Figures and Tables

**Figure 1 plants-10-02423-f001:**
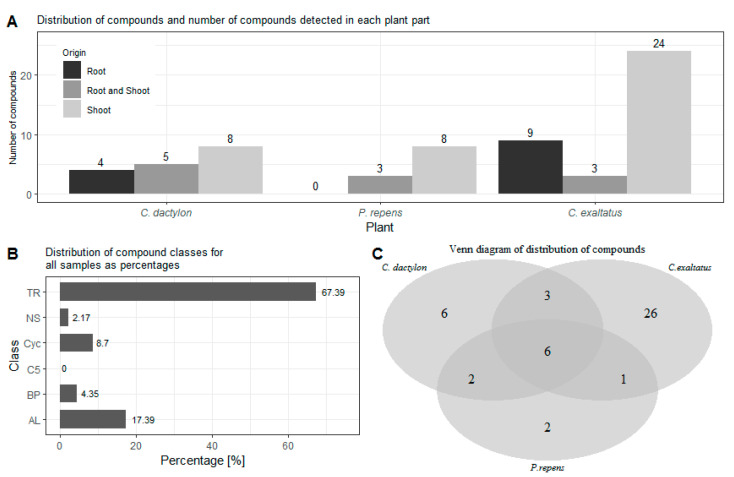
(**A**)—Shows the distribution of compounds detected in the roots, shoots, and both plant parts for each of the graminoid plants. (**B**)—Shows the percentage distribution of the classes for the identified compounds for all samples analyzed. (**C**)—Shows a Venn diagram of the number of unique compounds detected for each of the three plants as well as the number of compounds found in two or all three plant species. TR—Terpenoid, AL—Aliphatics, BP—Benzenoid/Phenylpropanoids, C5—C5-branched compounds, NS—Nitrogen- or sulfur-containing compounds, Cyc—Cyclic miscellaneous compounds.

**Figure 2 plants-10-02423-f002:**
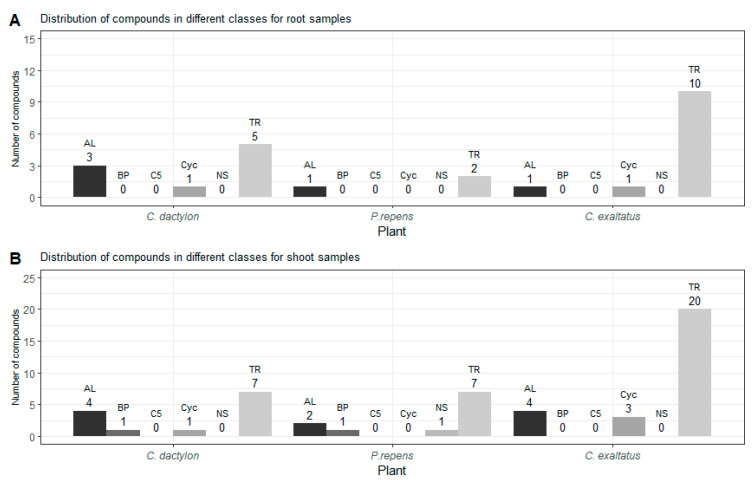
(**A**)—Shows the distribution of compounds for each class detected in the root samples for the three plants. (**B**)—Shows the distribution of compounds for each class detected in the shoot samples for the three plants. TR—Terpenoid, AL—Aliphatics, BP—Benzenoid/Phenylpropanoids, C5—C5-branched compounds, NS—Nitrogen- or sulfur-containing compounds, Cyc—Cyclic miscellaneous compounds.

**Table 1 plants-10-02423-t001:** Volatile compounds detected in the shoots and root parts of the graminoid plants *C. dactylon*, *C. exaltatus*, and *P. repens*.

S					Relative Composition % [Peak Area/Total Area] ± Standard Error					
					*C. dactylon*		*C. exaltatus*		*P. repens*	
No.	Compound	RT	RI	Class	Root	Shoot	Root	Shoot	Root	Shoot
	* Alcohols *									
1	2-octyn-1-ol	7.17	979	AL	-	0.47 ^a^	-	-	-	-
2	phenylethyl alcohol	9.60	1121	BP	-	-	-	-	-	8.72 ± 3.32
3	2,6-nonadien-1-ol	10.41	1171	AL	-	-	-	0.16 ^a^	-	-
**4**	**citronellol**	**11.37**	**1231**	**TR**	**-**	**2.51 ± 0.93**	**-**	**0.2 ^a^**	**-**	**5.35 ± 2.47**
	* Aldehydes *									
5	citral	8.94	1082	TR	-	-	-	0.51 ± 0.29	-	-
6	nonanal	9.23	1097	AL	1.37 ± 0.79	-	-	0.32 ± 0.19	-	-
7	2,6-nonadienal	10.19	1158	AL	-	-	-	0.57 ± 0.33	-	-
8	2-nonenal	10.29	1164	AL	-	-	-	1.25 ± 0.47	-	-
	* Amines *									
9	Myrtanylamine ^b^	9.52	1116	NS	-	-	-	-	-	0.43 ^a^
	* Aromatics *									
10	Mesitylene ^b^	7.53	998	BP	-	4.21 ± 1.46	-	-	-	-
	* Epoxides *									
**11**	**caryophyllene oxide**	**16.71**	**1609**	**TR**	**-**	**-**	**-**	**0.16 ^a^**	**-**	**-**
	* Furans *									
12	2-pentyl-furan	7.47	994	Cyc	4.04 ± 2.33	-	-	4.49 ± 1.18	-	-
13	2-(2-pentenyl)furan	7.63	1004	Cyc	-	-	-	3.2 ± 0.76	-	-
	* Hydrocarbons *									
**14**	**α-pinene**	**6.52**	**943**	**TR**	**8.03 ± 3.64**	**3.77 ± 1.35**	**8.73 ± 5.06**	**-**	**2.99 ^a^**	**14.42 ± 4.63**
**15**	**β-pinene**	**7.26**	**984**	**TR**	**9.34 ± 3.3**	**16.38 ± 1.89**	**16.69 ± 1.61**	**0.41 ± 0.26**	**38.7 ± 2.22**	**16.35 ± 2.45**
16	β-myrcene	7.46	994	TR	-	8.61 ± 1.98	0.83 ± 0.51	-	-	12.05 ± 1.85
17	decane	7.59	1001	AL	1.24 ± 0.74	3.3 ± 1.13	-	-	-	1.44 ± 0.95
**18**	**3-carene**	**7.84**	**1017**	**TR**	**-**	**-**	**0.73 ± 0.42**	**-**	**-**	**1.28 ± 0.75**
**19**	**limonene**	**8.15**	**1036**	**TR**	**2.96 ± 1.71**	**8.05 ± 1.6**	**1.99 ± 1.19**	**-**	**-**	**4.67 ± 2.7**
20	1-undecyne	8.81	1074	AL	-	0.88 ^a^	-	-	-	-
21	α-copaene	13.80	1392	TR	-	-	-	3.51 ± 0.25	-	-
22	β-cubebene	13.89	1398	TR	-	-	-	0.98 ± 0.34	-	-
23	β-elemene	14.01	1406	TR	-	-	-	22.92 ± 5.87	-	-
24	cyperene	14.22	1422	TR	-	-	10.34 ± 0.93	4.72 ± 0.33	-	-
25	γ-elemene	14.35	1432	TR	-	-	3.85 ± 3.02	-	-	-
26	α-bergamotene	14.38	1434	TR	1.17 ± 0.68	-	-	-	-	-
**27**	**α-cedrene**	**14.40**	**1435**	**TR**	**-**	**-**	**3.76 ± 1.55**	**-**	**-**	**-**
**28**	**caryophyllene**	**14.48**	**1441**	**TR**	**-**	**-**	**-**	**8.68 ± 0.44**	**-**	**-**
29	β-gurjunene	14.60	1450	TR	-	-	3.5 ± 2.02	-	-	-
30	α-gurjunene	14.83	1467	TR	-	-	-	0.17 ^a^	-	-
31	germacrene D	14.87	1469	TR	21.67 ± 7.05	-	-	-	-	-
**32**	**humulene**	**14.95**	**1475**	**TR**	**-**	**-**	**-**	**6.61 ± 0.4**	**-**	**-**
33	β-selinene	15.05	1483	TR	-	-	11.81 ± 4.3	1.33 ± 0.11	-	-
34	α-selinene	15.20	1493	TR	-	-	-	4.9 ± 0.25	-	-
35	valencene	15.26	1497	TR	-	-	-	1.17 ± 0.06	-	-
36	α-bulnesene	15.41	1509	TR	-	-	-	10.18 ± 1.86	-	-
37	α-farnesene	15.46	1512	TR	-	-	-	1.37 ± 0.12	-	-
38	4,11-selinadiene	15.52	1517	TR	-	-	-	9.22 ± 1.94	-	-
39	δ-cadinene	15.83	1542	TR	-	-	-	1.03 ± 0.36	-	-
40	bi-1-cycloocten-1-yl ^b^	15.89	1546	Cyc	-	-	1.2 ± 0.7	-	-	-
	* Ketones *									
41	3-octanone	7.35	988	AL	14.65 ± 1.21	11.01 ± 1.83	15.57 ± 2.94	-	9.95 ± 5.75	13.83 ± 4.25
42	sulcatone	7.35	989	TR	-	-	-	1.95 ± 0.38	-	-
43	2,2,6-trimethylcyclohexanone	8.24	1042	Cyc	-	0.91 ± 0.53	-	0.21 ^a^	-	-
44	α-isophorone	8.66	1066	TR	-	0.41^a^	-	-	-	-
45	geranyl acetone	14.70	1458	TR	-	-	-	0.76 ± 0.03	-	-
46	β-ionone	15.26	1497	TR	-	1.79 ± 1.04	-	-	-	1.26 ± 0.79

RT—Retention time [min], RI—Experimental retention index calculated using a C7-C40 alkane standard. TR—Terpenoid, AL—Aliphatics, BP—Benzenoid/Phenylpropanoids, C5—C5-branched compounds, NS—Nitrogen- or sulfur-containing compounds, Cyc—Cyclic miscellaneous compounds. Number of analyses for each column = 4. Compounds in bold font were identified with an external standard in addition to mass spectral identification and retention index. ^a^—Compound was only detected once in the 4 headspace sample collections; thus, no standard error can be reported. ^b^—Compounds have not been detected from a natural source in the current literature.

**Table 2 plants-10-02423-t002:** Post-hoc Tukey analysis of five compounds found in the headspace of at least two of the three graminoid plants.

			95 % Confidence Interval				
Compound	(I) Plant	(II) Plant	Mean Difference	Lower	Upper	*p*-Value	Significantly Different
α-pinene							
	*P. repens*	*C. dactylon*	6.058835	−2.923069	15.04074	0.161362	No
β -pinene							
	*P. repens*	*C. dactylon*	12.830001	0.4544081	25.20559	0.0414622	Yes
	*P. repens*	*C. exaltatus*	16.122119	3.2082078	29.03603	0.0134851	Yes
	*C. exaltatus*	*C. dactylon*	−3.292118	−16.5954810	10.01124	0.8048685	No
3-octanone							
	*P. repens*	*C. dactylon*	3.0250021	−4.699573	10.749577	0.5777641	No
	*P. repens*	*C. exaltatus*	0.2862853	−8.946348	9.518918	0.9964308	No
	*C. exaltatus*	*C. dactylon*	2.7387168	−6.020128	11.497562	0.7014438	No
β-myrcene							
	*P. repens*	*C. dactylon*	3.447685	−3.19232	10.08769	0.250943	No
citronellol							
	*P. repens*	*C. dactylon*	3.7893436	−3.091271	10.66996	0.2009792	No

**Table 3 plants-10-02423-t003:** Post-hoc Tukey analysis of six compounds found in the headspace samples of both the root and shoot samples of each of the three graminoid plants.

				95 % Confidence Interval				
Plant	Compound	(I) Part	(II) Part	Mean Difference	Lower	Upper	*p*-Value	Significantly Different
*C. dactylon*								
	α-pinene							
		Shoot	Root	−4.265665	−13.76537	5.234041	0.3140036	No
	β-pinene							
		Shoot	Root	7.038193	−2.267427	16.34381	0.113681	No
	3-octanone							
		Shoot	Root	−3.63632	−9.012898	1.740259	0.1490245	No
	decane							
		Shoot	Root	2.052726	−1.2501	5.355553	0.1791336	No
	limonene							
		Shoot	Root	5.093832	−0.63325	10.82091	0.0724245	No
*C. exaltatus*								
	β-pinene							
		Shoot	Root	−16.27886	−20.27341	−12.28431	0.0000588	Yes
	cyperene							
		Shoot	Root	−5.619737	−8.031604	−3.20787	0.0012586	Yes
	β-selinene							
		Shoot	Root	−10.47628	−21.00256	0.05001014	0.050796	No
*P. repens*								
	β-pinene							
		Shoot	Root	−22.34587	−30.42966	−14.26208	0.0005098	Yes
	3-octanone							
		Shoot	Root	3.887919	−13.6089	21.38473	0.6062301	No

## Data Availability

The data presented in this study are available on request from the corresponding author.

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
