# Peer review of "Chemical Diversity between Three Graminoid Plants Found in Western Kenya Analyzed by Headspace Solid-Phase Microextraction Gas Chromatography–Mass Spectrometry (HS-SPME-GC-MS)"

_plants, 2021, doi:10.3390/plants10112423_

Round 1

Reviewer 1 Report

The authors have made changes as suggested by reviewer.

Reviewer 2 Report

I thank the authors of the manuscript entitled “Chemical diversity between three graminoid plants found in Western Kenya analyzed by Headspace Solid-Phase Microextraction Gas Chromatography–Mass Spectrometry (HS-SPME-GC-MS)” for their reply to my comments.

I am aware of the problem of importing grass from Kenya to Sweden and it is clear to me that this work is related to that performed by Bokore et al. (I thank you for providing the complete reference of this work). However, I cannot find the novelty of the work you have submitted to Plants, especially compared to those done in collaboration with Bokore et al. In fact, in the aforementioned work a olfactometric assay was performed with whole live plants together with a dynamic head space sampling of the VOCs emitted by the plants and a subsequent GC-MS analysis. It is true that in the present work a different sampling method was adopted but, as I said in my previous revision, it was performed on altered plant parts, contrary to the other work in which the sampling was performed from intact live plants.

For this reasons, I would suggest to convert this work from “original research article” to “short communication”, if possible, after a major revision. I would change the aim of the work, I would focus more on the sampling technique, in my opinion you should describe the analysis of the selected plants by HS-SPME-GC-MS in comparison to the dynamic head space sampling and GC-MS analysis performed by Bokore et al.

In fact it is not true that you analyzed the different plants under the same conditions reported by Bokore et al., in that work the olfactometric assays were performed on whole live plants. Moreover, it is not true that the VOCs composition of these plants has never been investigated since it was done in the aforementioned work.

Some other considerations on the manuscript:

-did you optimize the HS-SPME sampling method?

-can you describe more in details the differences and similarities of the VOCs data obtained in this work compared to those obtained by Bokore et al? Which of the two adopted methods is more reliable and easy to handle?

Reviewer 3 Report

Manuscript ID: plants- 1431073        

Title: Chemical diversity between three graminoid plants found in Western Kenya analyzed by Headspace Solid-Phase Microextraction Gas Chromatography–Mass Spectrometry (HS-SPME-GC-MS)

Authors: Linus Svenberg *, Åsa Emmer

Submitted to section: Phytochemistry

The revised paper entitled “Chemical diversity between three graminoid plants found in Western Kenya analyzed by Headspace Solid-Phase Microextraction Gas Chromatography–Mass Spectrometry (HS-SPME-GC-MS)” fulfills all the recommendations of the reviewers. It has to be taken into account that the main goal of this study is to show the VOCs diversity between three graminoid plants, thus this issue is suitable for publishing in Plants. Moreover the authors explained their motivation for this investigation and discuss the hypothesis some of the volatile organic compounds to be possible oviposition attractants. This could be taken only as positive characteristic of the study; hence the obtained data provide basic practical information to test novel blends.

 I have found only one grammatical error in Table 1 is written 3-ctanone instead of 3-octanone.

Reviewer 4 Report

The authors improved their manuscript according to the reviewer's suggestions and comment and answered all the question raised throughout the revision.

Therefore, according to my opinion, the manuscript in this form can be accepted for publishing in Plants. 

Reviewer 5 Report

The authors improved the manuscript significantly, but there are some unclear points that the authors could improve in their review. The authors mention in the line 63-65 that “there is no gas phase chemical analysis performed on these graminoid plants from the region of Western Kenya” but in the manuscript of Bokore et al (Bokore, G. E. et al. Grass-like plants release general volatile cues attractive for gravid Anopheles gambiae s.s malaria vector mosquitoes. Preprint, doi:10.21203/rs.3.rs-645177/v1 (2021)) there is identification of VOC from the same plants from the same region. Which article provide the original data? I guess the authors should revise the text carefully.

The authors should revise the Discussion in order to avoid many conjectures, such as, “The larger number of compounds detected for this plant could be a basis for distinguishing the plants by family, as the more fragrant C. exaltatus belongs to the Cyperaceae family. Both C. dactylon and P. repens belong to the Poaceae family of plants, and when compared to that of C. exaltatus, has a similarly low number of uniquely detected compounds in their headspace.” A reference needed to such statement.

Minor points

“families” in the taxonomical classification should not be in italic

Line 182 and 184 - Anopheles arabiensis should be A. arabiensis

Line 267 and 269 – An. Arabiensis and An. gambieae should be A. arabiensis and A. gambiae

Reviewer 6 Report

In this revised version the authors improved the quality of the manuscript and then, in my opinion, it can be published

Round 2

Reviewer 2 Report

Dear Authors,

Thank you very much for your reply to my comments. I'm still not completely sure about the scientific soundness of this manuscript but, considering the changes made in the introduction, based on the previous comments, I think it is now more clear the aim of the work and the differences between this study and those performed by Bokore et al. I would suggest few minor revisions:

-you stated that the optimization of the HS-SPME sampling method was performed in a previous work, I would suggest to clarify it in the materials and methods by adding the reference

-I would add the comments included in your "response5" in the manuscript, in the results and discussion or in the conclusions sections.

After these minor revisions I would consider this manuscript accepted for publication in Plants.

Author Response

This manuscript is a resubmission of an earlier submission. The following is a list of the peer review reports and author responses from that submission.

Round 1

Reviewer 1 Report

In my opinion the manuscript can be published in Plants

The paper is well written; the results are clearly presented and discussed. The correlation between plants – compounds found is given. There are enough references, including up-to-date ones.   

However, the Table 1 is missing in the text, so is should be added.

Also, use the same font size through the text (row 171; 198-207 etc.)

In Table 2 – put the 95 % confidence Interval in the same raw

  • Add the line between p-value and Significantly different (as it is in Table 3)

Reviewer 2 Report

The present manuscript titled “Chemical diversity between three graminoid plants found in Western Kenya analyzed by Headspace Solid-Phase Microex-3 traction Gas Chromatography–Mass Spectrometry (HS-SPME-4 GC-MS)” reports the analysis of the volatile organic compounds (VOC) from three graminoid plants (Cynodon dactylon, Cyperus exaltatus and Panicum repens) with presumed ovopositional effect on the malaria vector Anopheles gambiae. The topic of the research is potentially interesting since malaria is an important disease spread in several countries. The monitoring of malaria vectors is important to contain its spread and the development of new outdoor-focused techniques is an interesting aspect. However, in my opinion, there is an inconsistency between the aim of the study and the method adopted in the analysis of the plant material. In fact, in the introduction (lines 79-83) the authors state that SPME can be adopted for the determination of VOCs in field experiment, in fact this sampling method can be applied in “in vivo” determination of VOCs, without altering the plant and its chemical composition. However, in this work the analyzed plants were previously collected, dried and further altered by the addition of water in the flask before the exposure to the fiber. The authors declare at lines 74-76 that manipulations of the plants can influence the overall composition of the volatiles, since the aim of this work is to evaluate the chemical composition of plant species compounds emitted in the environment, with a correlation to their potential ovipositional effect on A. gambiae, I think the sampling method adopted is not pertinent. Different in vivo sampling methods, such as dynamic headspace (D-HS), static headspace (S-HS) or direct contact (DC) methods in association with gas chromatography (GC) and mass spectrometry (MS) should be adopted. Moreover, the number of replicates for each species should be in a higher number, first of all different individuals should be collected for each species and analyzed individually, to verify a stability in the VOCs emission and potential differences among the considered species.

For these reasons, unfortunately, I would not considered the present manuscript for publication in Plants.

Reviewer 3 Report

Manuscript ID: plants-1399116

Title: Chemical diversity between three graminoid plants found in Western Kenya analyzed by Headspace Solid-Phase Microextraction Gas Chromatography–Mass Spectrometry (HS-SPME-GC-MS)

Authors: Linus Svenberg *, Åsa Emmer

Submitted to section: Phytochemistry

Comments to the manuscript

The paper entitled “Chemical diversity between three graminoid plants found in Western Kenya analyzed by Headspace Solid-Phase Microextraction Gas Chromatography–Mass Spectrometry (HS-SPME-GC-MS)” explores an excellent designed, very important investigation directed to find of what compounds could constitute a starting point for novel blends of volatile organic compounds to be tested as oviposition attractants. Thus, HS-SPME GC-MS analysis was performed on shoots and root parts from the three graminoid plants Cynodon dactylon, Cyperus exaltatus, and Panicum repens to determine the composition of volatile compounds acquired from each plant.

Materials and Methods are appropriately described, as well as, the identification of VOCs. The term VOCs should be used through the whole manuscript, instead of VOC in 225 row and elsewhere (44 row). Some corrections of the font format in row 178, between 198 row and 206-7 are also needed.

I recommend to showing Table 1 in the manuscript and not in the Supplementary Materials.

Reviewer 4 Report

Overall:

Reading the introduction, this manuscript seemed to be very interesting showing some new data which can be very applicable in changing the oviposition behavior of Anopheles. However, the rest of the manuscript is rather disappointing.

Namely, there is only one method used HS-SPME in a combination with GC-MS to determine the presence of the compounds in three Western Kenya plants. Later, the 1-way ANOVA was performed to compare the composition of three plants and suggest the potential compound which could be used as a novel blend.

The authors’ conclusion is that the searching for that compound “could be based on the compounds reported for C. dactylon and C. exaltatus based on the comparisons performed for the three graminoid plants”.  They don’t suggest which compound, which combination of compounds etc and, therefore, the novelty and the scientific significance is rather poor.

Here are some of the specific comments:

Keywords:

Cynodon dactylon, Cyperus exaltatus, Panicum repens - should be in italics

Results and Discussion:

Table 1 is fully described in the results section, but it can be found in the supplementary file.

It should be called supplementary table 1 or put the table 1 below the tittle.

When looked in the excel file (in supplementary) in sheet entitled Table 1, there is a text box with description of Table 2?

Figure 1 is of bad quality. It should be of better resolution. It would be better to write Both root and shoot on 1A instead of just Both. Also, explain abbreviation of the compound classes in the figure legend. Name of the plants in italics.

Figure 2. same as previous - It should be of better resolution. Explain abbreviation of the compound classes in the figure legend. Name of the plants in italics.

Why there are tables 2 and 3 both in the main manuscript and in the supplementary data?

Materials and Methods:

Line 316: there is not such a figure in the supplementary file

Conclusions:

The conclusions that “there was no significant difference in the total relative peak areas detected in shoot and root samples for any of the overlapping compounds except for β-pinene. The relative peak area for this compound was significantly higher in P. repens compared to the other two plants, while no significant difference was shown for this substance between C. dactylon and C. exaltatus.” and furthermore “It is suggested that a starting point for investigations of novel blends for ovipositional attractive combinations of VOCs could be based on the compounds reported for C. dactylon and C. exaltatus based on the comparisons performed for the three graminoid plants.” Are in the contrary. What compounds should be tested for novel blends? What is the application of this study?

Minor corrections:

Some typos:

  • Add spaces before reference
  • Line 46-47: do you miss a verb in the first part of the sentence
  • Latin plant or insect name should always be in italics
  • Line 132-135: missing verb in the sentence

Line 58: explain the abbreviation HS-SPME (first time appearing in the text; beside the abstract)

Classes of compounds sometimes written with capital first letter, and sometimes with a small one

Reviewer 5 Report

The authors presented a manuscript in which they analyze volatile organic compounds (VOC) from three graminoids from Western Kenya using HS-SPME-GC-MS and suggest that these VOC may attract Anopheles females to oviposition. Reading the manuscript there is some evidence that these VOC can attract Anopheles for oviposition, but there are no conclusive results as described in Boroke et al, 2020. It seems that there are more results in the reference “Bokore, G. E. et al. (eds Thomas Odhiambo Campus International Centre of Insect Physiology and Ecology & Royal 450 Institute of Technology) (2021).”, but I could not find the reference. Up to now, only cedrol is considered as Anopheles attractant (Lindh, J. M. et al. Discovery of an oviposition attractant for gravid malaria vectors of the Anopheles gambiae species complex. Malar J 14, 119 (2015)).

Since malaria is an important disease in tropical countries, an appropriate way to control the vector (Anopheles mosquitoes) proliferation is welcome. But the rationale and the procedures to collect VOC data of this manuscript seem not appropriate. The authors collected VOC data shipped from Kenya and then froze the material to collect the samples. In my opinion, to get any information about semiochemicals, the collection should be done in the field and from fresh material. Considering this point, I would like to address some questions.

  • Can the authors confirm that the VOC collected are equivalent to those emitted by plants in the field?
  • The first question is not only related to the number of the compounds, but also to the proportion between compounds that is crucial to attraction behavior.
  • The authors used only shoots and rhizomes. Is there any reason for that? Why are they not collected from leaves?
  • Why did the authors not joined these results to this manuscript “Grass-Like Plants Release General Volatile Cues Attractive for Gravid Anopheles Gambiae S.S. Malaria Vector Mosquitoes” published in the Research Square?

Considering these points, the manuscript has several inappropriate methods and information to correlate these VOCs with attraction for oviposition by Anopheles females.

Reviewer 6 Report

The authors present a manuscript entitled "Chemical diversity between three graminoid plants found in 2 Western Kenya analyzed by Headspace Solid-Phase Microex- 3 traction Gas Chromatography–Mass Spectrometry (HS-SPME- 4 GC-MS)". The work is well written and organized, the topic is of great interest to Plants readers. The study of the VOCs is presented correctly and the experiments are clearly described so that they can be easily reproduced. The results are expressed correctly and the conclusions are consistent with the purpose of the work. As minor issues authors should check the punctuation and formatting of references in the text. Furthermore, all botanical names must be written in italics. As a suggestion to complete the excellent work, the authors could propose the characteristic volatile compounds as qualitative markers useful to identify the species that has the reported potential.